# Data-Efficient Training with Physics-Enhanced Deep Surrogates

**Raphaël Pestourie,**[1] **Youssef Mroueh,** [2,3] **Chris Rackauckas**[1] **Payel Das** [2,3] **Steven G. Johnson** [1]

[1] MIT, 77 Massachusetts Ave, Cambridge, MA 02139, USA
[2] IBM Research AI, IBM Thomas J Watson Research Center, Yorktown Heights, NY 10598, USA
[3] MIT-IBM Watson AI Lab, Cambridge, MA 02139, USA
rpestour@mit.edu

## Abstract

We present a "physics-enhanced deep-surrogate" ("PEDS") approach to fast surrogate models for complex physical systems described by partial differential equations (PDEs) and similar models: we embed a low-fidelity "coarse" solver layer in a neural network that generates "coarsified" inputs, trained end-to-end to globally match the output of an expensive high-fidelity numerical solver. In this way, by incorporating complex physical knowledge in the form of the low-fidelity model, we find that a PEDS surrogate can be trained with at least 10× less data than a "black-box" neural network for the same accuracy. Asymptotically, PEDS appears to learn with a steeper power law than black-box surrogates, and benefits even further in combination with active learning. We demonstrate this using an example problem in electromagnetic scattering that appears in the large-scale optimization of optical metamaterials using scientific computing.

## Introduction

In mechanics, optics, thermal transport, physical chemistry, climate models, and many other fields, data-driven surrogate models—such as polynomial fits, radial basis functions, or neural networks—are widely used as an efficient solution to replace repetitive calls to slow numerical solvers (Baker et al. 2019; Benner, Gugercin, and Willcox 2015; Willard et al. 2020; Hoffmann et al. 2019; Pestourie et al. 2018). However the reuse benefit of surrogate models comes at a significant cost in training time, where a costly high-fidelity numerical solver must be evaluated many times to provide an adequate training set, and this cost rapidly increases with the number of model parameters (the "curse of dimensionality") (Boyd 2001). In this paper, we explore one promising route to increasing training-data efficiency: incorporating *some* knowledge of the underlying physics into the surrogate by training a generative neural network (NN) "end-to-end" with an *approximate* physics model. We call this hybrid system a "physics-enhanced deep surrogate" (PEDS), and demonstrate multiple-order-of-magnitude improvements in sample and time complexity on a test problem involving optical metamaterials—composite materials whose properties are designed via microstructured geometries (Pestourie et al. 2020). In inverse design of metamaterials, similar geometric

components may be re-used thousands or millions of times in a large structure such as an optical metasurface (Pestourie et al. 2018; Pestourie 2020), making surrogate models especially attractive to accelerate computational design (Bayati et al. 2021; Li et al. 2021).

In particular, we present a PEDS architecture for modeling transmission through a microstructured multilayer "metasurface" (Pestourie et al. 2020), where the high-fidelity model solves Maxwell's equations, in which a deep NN is combined with an fast approximate Maxwell solver based on an extremely coarse discretization, as depicted in Fig. 1 (Sec. ). By itself, the coarse model yields $> 100\%$ error if it is applied directly to a downsampled/coarsified geometry, but it qualitatively captures key underlying physics of scattering and resonance. To obtain an accurate surrogate, we apply a deep NN to *generate a coarse geometry*, *adaptively mixed* with the downsampled input geometry, which is then used as an input into approximate solver and trained end-to-end to minimize the overall error. In this way, the NN learns to nonlinearly correct for the errors in the coarse model, but at the same time the coarse model "builds in" some knowledge of the physics and geometry. We compare the result of our PEDS model against a NN-only baseline model (Sec. ) as well as previous "space-mapping" (SM) (Bakr et al. 2000; Zhu et al. 2016; Feng et al. 2019) approach where we combine a coarse Maxwell solver with a NN transforming only a low-dimensional parameterization of the fine geometry to a similar low-dimensional parameterization of the coarse geometry (Sec. ). We find that PEDS not only lowers the error of the surrogate for a given amount of data, but it actually seems to improve the asymptotic *rate* of learning ($\approx 5\times$ larger power law), so that the benefits increase as accuracy tolerance is lowered (Fig. 2 and Sec. ). For 3.5% accuracy, PEDS requires several orders of magnitude less data than the competing approaches. We show through an ablation study that adding information from the downsampled structure increases the accuracy by 15% in a low-data regime. Furthermore, we find that PEDS gains significant additional benefits by combining it with active-learning techniques from our earlier work (Pestourie et al. 2020), and in fact the benefits of active learning seem to be even greater for PEDS than for competing approaches. Although the resulting PEDS surrogate is more expensive to evaluate than a NN by itself, due to the coarse Maxwell

solver, it is still much faster than the high-fidelity Maxwell solver ($> 100\times$ in 2D, $> 10^4\times$ in 3D). Furthermore, since our NN generates a coarsified version of the geometry, this output can be further examined to gain insight into the fundamental physical processes affecting the output.

PEDS should not be confused with physics-informed neural networks (PINNs), which solve the full PDE (imposed pointwise throughout the domain) for the entire PDE solution (*not* a surrogate for a finite set of outputs like the complex transmission) (Karniadakis et al. 2021; Lu et al. 2021b), and which do not employ any pre-existing solver; indeed, current PINNs tend to be slower than conventional "fine" PDE solvers (e.g. based on finite elements) (Shin, Darbon, and Karniadakis 2020), but offer potentially greater flexibility. Universal ordinary differential equations (UODEs) (Rackauckas et al. 2020) also tackle a different problem from PEDS: they identify unknown dynamics in an ODE by replacing the unknown terms with neural networks trained on data. In contrast to DeepONet (Lu et al. 2021a) and Fourier neural operators (Li et al. 2020), PEDS includes a numerical solver layer. Finally, in contrast to error correction techniques at the output level of the surrogate (Lu et al. 2020; Koziel, Bandler, and Madsen 2006), PEDS includes the solver in an end-to-end fashion during the training process.

## Results

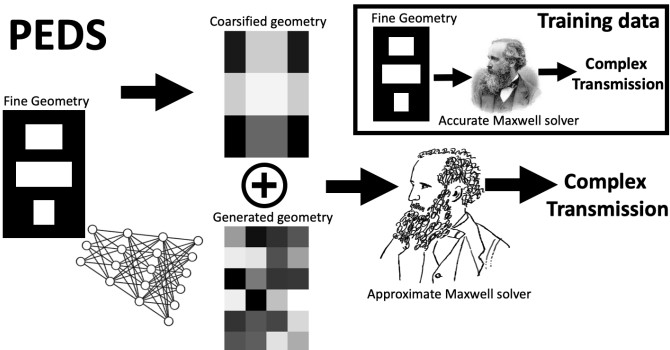

Figure 1: Diagram of PEDS: (Main) from the geometry parameterization, the surrogate generates a coarse structure which is combined with a downsampled version of the geometry (in a pixel averaging sense) to be fed in a coarse solver for Maxwell's equations (symbolized by a cartoon picture of James Clerk Maxwell). (Inset) the training date is generated by solving more costly simulations directly on a fine solver (symbolized by a photograph of James Clerk Maxwell).

## Physical model and solvers

Similarly to (Pestourie et al. 2020), our surrogate model predicts the complex transmission $t^{fine}(p)$ of a 2D "meta-atom" unit cell with a parameterized geometry $p$, which consists of ten layers of air holes with independent widths etched in a substrate (of dielectric constant $\varepsilon = 2.1$ corresponding to silica), with periodic boundary conditions in $x$

and outgoing radiation boundary conditions in the $y$ direction and an incoming normal-incident planewave from below. In terms of the vacuum wavelength $\lambda$ of the incident wave (for the largest $\lambda$ considered below), the period in $x$ is $0.95\lambda$ and the total thickness is $11\lambda$ (with hole heights of $0.75\lambda$ and interstices of $0.35\lambda$); the fact that the structure is several wavelengths in diameter causes the transmission $t^{fine}(p)$ to be a complicated oscillatory function that makes the surrogate training challenging (Pestourie et al. 2020). (A "metasurface" consists of a collection of many of these meta-atoms, designed to perform some optical function such as focusing (Li et al. 2021). The full solution for a metasurface can be approximated in terms of the transmissions of individual periodic 'unit cells via a local periodic approximation (Pestourie et al. 2018).) A schematic unit cell with 3 holes is showed in Fig. 1, and an example 10-hole structure from the training set is shown in Fig. 2 (right).

Both the "fine" (high-fidelity) and "coarse" (low-fidelity) solvers in this paper employ finite-difference frequency-domain (FDFD) discretizations of Maxwell's equations (Champagne II, Berryman, and Buettner 2001), using perfectly matched layers (PMLs) (Sacks et al. 1995) to implement outgoing boundary conditions. FDFD essentially represents the geometry by a grid of discretized $\varepsilon$ "pixels," which is some function fine($p$) of the parameters (hole widths) $p$. (In particular, each pixel's $\varepsilon$ is assigned to a subpixel average of the infinite-resolution structure, which both increases accuracy (Oskooi, Kottke, and Johnson 2009) and makes fine($p$) piecewise differentiable.)

An FDFD resolution of 40 pixels per wavelength is used as our "fine" solver, the source of our training data as indicated in Fig. 1 (inset). This resolution is typical for high-fidelity solvers in electromagnetism, because it is comparable to the manufacturing accuracy in nanophotonics and hence suffices for practical metalens design (Li et al. 2021; Bayati et al. 2021) within fabrication uncertainty. (Sharp/narrowband resonances can shift if one refines the resolution further, but the positions and the bandwidths of the resonances are accurate to within a few percent.) Each fine-solver data point required $\approx 1$ s (on a 3.5 GHz 6-Core Intel Xeon E5); an analogous simulation in 3D takes tens of minutes. Our PEDS surrogate (below) uses an FDFD solver at a coarser resolution of 10 pixels per wavelength, which is about $100\times$ faster in 2D and $> 10^4\times$ faster in 3D, but has much worse accuracy; as quantified in Sec. , it differs from the fine solver's transmission by $> 100\%$ on our test set.

The upfront cost of building the training dataset $S = \{(p_i, t_i^{fine}), i = 1...N\}$ is the most time-consuming part of developing a supervised surrogate model. By building some approximate "coarse" physics knowledge into the surrogate, we will show that PEDS greatly reduces the number $N$ of expensive simulations, especially when combined with active learning.

## PEDS

The PEDS surrogate model $\tilde{t}(p)$ is shown schematically in Fig. 1, and is computed in the following stages ("layers"):

1. Given the parameters $p$ of the geometry, a deep genera-

tive NN model yields a grid of pixels describing a coarse FDFD geometry. We call this function $\text{generator}_{\text{NN}}(p)$

2. We also compute a coarse-grid downsampling (sub-pixel averaging) of the geometry, denoted $\text{coarse}(p)$, e.g. by downsampling $\text{fine}(p)$.

3. We make a weighted combination $G$ of the NN-generated and downsampled geometries: $G(p) = w \cdot \text{generator}_{\text{NN}}(p) + (1 - w) \cdot \text{coarse}(p)$, with a weight $w \in [0, 1]$ (independent of $p$) sta that is another learned parameter.

4. If there are any additional constraints/symmetries that the physical problem imposes on the geometry, they can be applied as projections $P[G]$. (For example, in our metasurface problem we could average $G$ with its mirror image to ensure that the generated structure is mirror-symmetric like the exact structure.)

5. Finally, given $G$, we evaluate the coarse solver to obtain the complex transmission $\tilde{t}(p) = t^{\text{coarse}}(P[G(p)])$.

In summary, the PEDS model is

$$\tilde{t}(p) = t^{\text{coarse}}\left(P\left[w \cdot \text{generator}_{\text{NN}}(p) + (1-w) \cdot \text{coarse}(p)\right]\right)$$
(1)

A basic PEDS training strategy could simply minimize the mean-square error $\sum_{(p,t^{\text{fine}}) \in S} |\tilde{t}(p) - t^{\text{fine}}|^2$ (for a training set $S$) with respect to the parameters of the NN and the weight $w$. In our case, we employ a more complicated loss function that allows us to easily incorporate active-learning strategies (Pestourie et al. 2020). We optimize the Gaussian negative log-likelihood of a Bayesian model (Lakshminarayanan, Pritzel, and Blundell 2016)

$$- \sum_{(p_i, t_i^{fine}) \in S} \log \mathrm{P}_\Theta(t_i^{fine}|p_i) \propto$$

$$\sum_{(p_i, t_i^{fine}) \in S} \left[\log \sigma(p_i) + \frac{(t_i^{fine} - \tilde{t}(p_i))^2}{2\sigma(p_i)^2}\right]$$

where $\mathrm{P}_\Theta$ is a Gaussian likelihood parameterized by the model parameters $\Theta$, and the heteroskedastic "standard deviation" $\sigma(p) > 0$ is the output of another NN (trained along with our surrogate model). In practice, rather than examining the entire training set $S$ at each training step, we follow the standard "batch" approach (Goodfellow, Bengio, and Courville 2016) of sampling a random subset of $S$ and minimizing the expected loss with the Adam stochastic gradient-descent algorithm (Kingma and Ba 2014) (via the Flux.jl (Innes 2018b) software in the Julia language).

We train our model to predict the complex transmission for 3 frequencies, which are encoded with a one-hot encoding vector. The input of the model $p$ is the concatenation of the 10 widths and the one-hot encoding of the frequency. The coarse solver is a layer of the PEDS model, which is trained end-to-end, so we must backpropagate its gradient $\nabla_G t^{\text{coarse}}$ through the other layers to obtain the overall sensitivities of the loss function. This is accomplished efficiently using well-known "adjoint" methods (Molesky et al. 2018), which yield a vector-Jacobian product that is then automatically composed with the other layers using automatic differentiation (AD) (via the Zygote.jl (Innes 2018a) software).

**Static and dynamic training** In this paper, we investigated two types of supervised end-to-end training, a static training which takes a training set sampled at random, and a dynamic Bayesian training where the training set is iteratively expanded using an active learning algorithm (Pestourie et al. 2020). Essentially, active learning attempts to sample training points where the model uncertainty is greatest, thereby reducing the number of costly training points that must be generated (by the fine solver). Our previous work on active learning reached more than an order of magnitude improvement in data efficiency for a black-box NN, and in this paper (Sec. below) we also substantial improvements from active learning for PEDS.

The active-learning algorithm iteratively builds a training set by filtering randomly generated points with respect to a trained measure of uncertainty (Pestourie et al. 2020). The hyperparameters of this algorithm are $n_{\text{init}}$, which is the number of points the surrogate models is initially trained with, $T$, the number of exploration iteration, $M$ and $K$, which are such that $M \times K$ points are randomly generated at each iteration and only $K$ points with highest uncertainty $\sigma(p)$ are explored, i.e. we run the expensive fine solver to get the PDE solutions of these points. We have trained surrogates as well as *ensemble* of 5 independent surrogates. We found that models optimizing the negative log-likelihood perform similarly to models optimizing the mean squared error in the case static training. This is not surprising, because the mean squared error is part of the negative log-likelihood objective.

**SM baseline** Our PEDS has similarities with input space mapping (SM) (Koziel, Cheng, and Bandler 2008), especially neural SM (Bakr et al. 2000) and coarse mesh/fine SM (Feng et al. 2019), where the input of a fine solver is mapped into the input of a coarse solver. However, SM uses the same parameterization $p$ (e.g. the widths of the holes) for the fine solver and the coarse solver, whereas PEDS uses a much richer coarse-geometry input (a grid of material values, whose dimensionality is different in the coarse and fine geometries) and can therefore incorporate more geometric and physical inductive biases, such as symmetries and the downsampled structure. For comparison, we trained an input SM baseline model, which is a combination of neural and coarse mesh/fine mesh SM (Zhu et al. 2016; Feng et al. 2019). In this model, the NN is learning a mapping that creates modified geometry parameters $p$ which are then fed to the coarse solver (see Sec. for implementation details). Since the coarsified parameterized geometry is implemented via sub-pixel averaging as described in Sec. , this function is differentiable, so the gradient can backpropagate all the way to the mapping NN.

## Accuracy and performance

We compared PEDS to a NN-only baseline (Sec. ) and an SM baseline. In Fig. 2, we show that PEDS clearly outperforms all other models when combined with active learning.

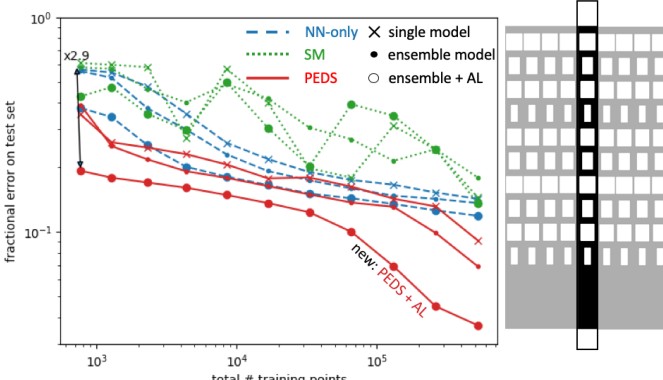

Figure 2: (Left) Fractional error (FE) on the test set: PEDS outperforms the other models significantly when combined with active learning (AL). SM performs poorly compared to PEDS and does not gain much accuracy from ensembling nor from active learning. (Right) Geometry of the unit cell of the surrogate model. Each of the 10 air holes have independent widths, the simulation is performed with periodic boundary conditions on the long sides, the incident light comes from the bottom and the complex transmission is measured at the top of the geometry.

In low-data regime, it is $2.9\times$ more accurate than the baseline. Asymptotically, in high-data regime, it converges to the true value with a power law exponent $5\times$ better, with a slope of -0.5, in contrast to -0.1 for the baseline on the loglog plot. SM does not gain much accuracy from ensembling nor from active learning, and is even worse than the baseline NN. (We found that SM can perform comparably to the baseline NN if the coarse-solver resolution is doubled to 20, not shown here, at the expense of $\approx 10\times$ more computational effort.)

From a data-efficiency perspective, the PEDS+AL solver achieves 20% accuracy on the test set with $\approx 21\times$ less data than the baseline NN, $\approx 8\times$ less data than the baseline NN with AL, and orders of magnitudes less data than space mapping (SM) with AL. Only PEDS+AL reaches 3.5% accuracy, but if we extrapolate the other curves it is clear that they would require at *least* two orders of magnitude more data to achieve similar accuracy.

Timing was compared on a 3.5 GHz 6-Core Intel Xeon E5. Evaluating the baseline (with an ensemble of neural networks) takes $500\ \mu s$, while PEDS evaluates in $5\ ms$, which is about a ten times slower. However the fine solver is about a hundred times slower, evaluating at $\approx 1s$. In order to simulate the data set quickly, and without loss of generality, we showed results for PEDS in 2D. Although PEDS is already faster than the fine model by two orders of magnitude, this difference will be even starker for 3D simulations. The simulation of the equivalent structure in 3D evaluates in about $100\ ms$ with the coarse model, and in $2462\ s$ with the fine model. In this occurrence, PEDS would represent a speed-up by at least four orders of magnitude. Moreover, as we discuss in Sec. , the general PEDS approach can be applied

to a wide variety of "coarse physics" models, which can be chosen to have a wide range of performance benefits compared to a high-fidelity model.

In order to understand the effect of mixing the generated structure with a downsampled structure, we performed an ablation study on an AL ensemble model in the low-data regime (1280 training points), with results given in Table 1. The edge cases of using only the downsampled structure with the coarse solver performs the worst (1.24 error), corresponding to $w = 0.0$ in Eq. (1). Conversely, using the NN generator only, corresponding to $w = 1.0$ in Eq. (1), is still about 15% worse (0.20 error) than using adaptive mixing $0 < w < 1$. Imposing mirror symmetry, via $P[G] = (G + \text{mirror image})/2$ in Eq. (1), did not improve the accuracy of the model in this case (but is a useful option in general, since symmetry may have a larger effect on the physics in other applications).

| Generative model for coarse geometry | FE on test set |
|---|---|
| $w = 0.0$ (coarsified only) | 1.24 |
| $w = 1.0$ (generator only) | 0.20 |
| PEDS with symmetry | 0.18 |
| PEDS | 0.17 |

Table 1: Ablation study of PEDS with ensembling and active learning for 1280 training points, showing the impact of mixing generated and coarsified geometries, as well of as imposing symmetry.

## Discussion and outlook

The significance of the PEDS approach is that it can easily be applied to a wide variety of physical systems. It is common across many disciplines to have models at varying levels of fidelity, whether they simply differ in spatial resolution (as in this paper) or in the types of physical processes they incorporate. For example, in fluid mechanics the "coarse" model could be Stokes flow (neglecting inertia), while the "fine" model might be a full Navier–Stokes model (vastly more expensive to simulate) (Ferziger, Perić, and Street 2002), with generator NN correcting for the deficiencies of the simpler model.

In addition to applying the PEDS approach to additional physical systems, there are a number of other possible technical refinements. For example, one could easily extend the PEDS NN to take an image of the fine-structure geometry rather than its parameterization, perhaps employing convolutional neural networks to represent a translation-independent "coarsification." Another interesting direction might be to develop new "coarsified" physics models that admit ultra-fast solvers but are too inaccurate to be used *except* with PEDS; for instance, mapping Maxwell's equations in 3D onto a simpler (scalar-like) wave equation or mapping the materials into objects that admit especially efficient solvers (such as impedance surfaces (Pérez-Arancibia, Pestourie, and Johnson 2018) or compact objects for surface-integral equation methods (Jin 2015)).

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

# Appendix

**Coarse solver and gradient** In the present work, the coarse solver is similar to the fine solver except that it uses a much coarser resolution of 10, which corresponds to a resolution of less than 5 pixels per wavelength in the worst case, instead of 40 for the fine model. The symmetry action was a simple mirror symmetry, implemented by averaging of the geometry with its mirror flip.

**Implementation details of PEDS and baselines** The generator neural network of PEDS has two hidden layers with 256 nodes and relu activation functions, and outputs a flattened version of the coarse geometry of dimension 1100 with a hardtanh activation function ($hardtan(x) = max(min(x, 1), 0)$). The network that outputs the variance of the models, takes the generated coarse geometry as input, has 3 hidden layers with relu activation functions and outputs a scalar with a relu activation function. The corresponding baseline, which is a neural-network only (NN-only) method, was chosen to be as close as possible to PEDS architecture, it replaces the coarse solver with a fully connected layer, and outputs two scalars with a tanh activation function. Note that it does not have the information of the downsampled structure. The mapping neural network of the input SM implementation has two hidden layers with 256 nodes and relu activation functions, and outputs the coarse geometry parameters of dimension 10 with a hardtanh activation function. The variance network is similar to PEDS except that the inputs are the SM output geometry parameters. The batch size was set to 64 and the learning rate to $10^{-3}$. Every training went through 10 epochs.

**Active learning implementation details** The active learning training (Pestourie et al. 2020) used the following parameters $n_{init} = 256$, $T = 8$, $M = 4$, and K took powers of 2 ranging from $2^6$ to $2^{16}$.

**Parallelization** In order to accelerate the training of the surrogate model, we parallelized the training at the batch loop level. For ensemble learning, we used 320 computing units which were split into 5 groups (one group per model in the ensemble) of 64 computing units. With a batch size of 64, each worker evaluates the surrogate only once per batch loop (a batch size of a multiple of 64 would work well too).

**Code** The code was implemented in Julia language version 1.6, using MPI.jl for parallelization with MPI, Flux.jl for the neural network training framework, ChainRules.jl for custom differentiation rules, and Zygote.jl for other automatic differentiation.