# OpenReview forum: "Data-Efficient Training with Physics-Enhanced Deep Surrogates"
_AAAI.org/2022/Workshop/ADAM — AAAI 2022 Workshop ADAM_

### Official Review · Reviewer_sSYZ · 2021-11-29
**Coarse-to-fine physics informed mapping, promising results**

**Rating:** 6
**Confidence:** 4

**Review:**

This paper presents a method (PEDS) to construct physics informed surrogate model using a coarse to fine framework where a NN is used to map the fine input to a 'generated' coarse input which is then combined with a coarse version of the fine input and fed to a coarse physics based solver. The NN + combination weights is trained end-to-end with the coarse solver. The method, though similar to neural and coarse-to-fine space mapping methods (except for combination of generated coarse and coarse version of input, which can have different dimensionalities), is shown to perform significantly better, especially when coupled with active learning.

The results would be strengthened if the comparison to SM was with a dimension 1100, the same dimension that is learnt by the NN since PEDS with generator only gets pretty close to PEDS as per Table 1 (perhaps within error bars?), so the improvement could just be larger embedding dimension. Also, section numbers are missing in reference to appendix.

---

### Official Review · Reviewer_jTZ1 · 2021-12-01
**Nice new method for developing deep surrogate PDE solvers in a data-efficient manner**

**Rating:** 7
**Confidence:** 4

**Review:**

The paper proposes a natural approach for improving sample (data) efficiency in deep surrogate PDE solvers. The high level idea is to use two spatial scales (first train a "low-fidelity" model and use its outputs as side information for a final, high fidelity model); this in combination with an active learning strategy leads to data reductions of upto 10x.

The paper is nicely written and the topic is timely.

Points of feedback: a) consider validating on more challenging complex systems. b) why only 2 scales? is there room for an extension to a hierarchy of coarse-to-fine mappings (analogous to multiresolution analysis).